# Pose Estimation of a Cobot Implemented on a Small AI-Powered Computing System and a Stereo Camera for Precision Evaluation

**DOI:** 10.3390/biomimetics9100610

**Published:** 2024-10-09

**Authors:** Marco-Antonio Cabrera-Rufino, Juan-Manuel Ramos-Arreguín, Marco-Antonio Aceves-Fernandez, Efren Gorrostieta-Hurtado, Jesus-Carlos Pedraza-Ortega, Juvenal Rodríguez-Resendiz

**Affiliations:** Facultad de Ingeniería, Universidad Autónoma de Querétaro, Cerro de las Campanas, Las Campanas, Queretaro 76010, Mexico; marco.aceves@uaq.mx (M.-A.A.-F.); efren.gorrostieta@uaq.mx (E.G.-H.); carlos.pedraza@uaq.mx (J.-C.P.-O.); juvenal@uaq.edu.mx (J.R.-R.)

**Keywords:** precision degradation, robot arm, robotic vision, stereo camera, convolutional neural network, 3D vision

## Abstract

The precision of robotic manipulators in the industrial or medical field is very important, especially when it comes to repetitive or exhaustive tasks. Geometric deformations are the most common in this field. For this reason, new robotic vision techniques have been proposed, including 3D methods that made it possible to determine the geometric distances between the parts of a robotic manipulator. The aim of this work is to measure the angular position of a robotic arm with six degrees of freedom. For this purpose, a stereo camera and a convolutional neural network algorithm are used to reduce the degradation of precision caused by geometric errors. This method is not intended to replace encoders, but to enhance accuracy by compensating for degradation through an intelligent visual measurement system. The camera is tested and the accuracy is about one millimeter. The implementation of this method leads to better results than traditional and simple neural network methods.

## 1. Introduction

Robotic vision and intelligent algorithms are of great importance for robotic manipulators, as they enable more accurate and precise task execution. As a result, the use of these methods has begun to find applications in metrology. Robotic manipulators are indispensable tools for the automotive or aerospace industries. Among other things, they perform repetitive, complex, or strenuous tasks, so robots become increasingly inaccurate and imprecise over time. Normally, the measurement of a degree-of-freedom (DoF), i.e., the number of independent joints or axes of movement that give it the ability to move, is achieved in angular units (degrees or radians). Sensors such as encoders, tachometers, gyroscopes, and others are used for this purpose.

A robotic manipulator is a mechanically articulated arm consisting of links connected by joints that allow relative motion between two consecutive links. The number of independent or minimum coordinates required to describe the robot position defines the number of DoF. Robotic arms move and manipulate tools called end effectors. These can be grippers, welding wires, drills, suction cups, cameras, or others. Figure 1 shows a prototype robot arm with 3 DoF, where (*x*, *y*, *z*) is the position of the end effector and $\theta_{1}$, $\theta_{2}$ and $\theta_{3}$ are the angular position of each DoF. The robot Tool Center Point (TCP) is the point used for positioning the robot in any robot program where targets are defined in Cartesian space. TCP is defined as a transformation from the robot mounting flange [1].

In the field of robotics, the precision of the robot is to reach a certain point in the working area with a minimum position error. Precision helps in evaluating the performance of a robot, and its absence is called precision degradation. It is divided into three groups, which are caused by structural deformations of the joints or geometric errors. Then there are the non-geometric errors such as friction and gear backlash. And, finally, the errors caused by computational processing. Typical geometric errors include tolerances, elasticities, eccentricities, and wear [2,3,4,5].

On the path to precision degradation analysis, computer vision devices and artificial intelligent techniques are methodologies implemented in robotic applications [6], such as control [7,8,9,10,11], inverse kinematics [12,13], and object segmentation [14,15,16]. Stereo cameras have become more precise in the last ten years [14,17,18,19,20], and actual microcomputers like Raspberry Pi model 4 or Jetson Nano 280 are powerful enough to perform real-time measuring tasks. However, there is little background on the use of these methods as an alternative measurement system [21], and even less on the specific use of stereo cameras and convolutional neural networks (CNNs), which are a type of deep learning (DL) model particularly effective for image analysis [22].

Table 1 contains state-of-the-art for this work. It has the reference of the author, the methodology, and improvements.

We consider that, given the current state-of-the-art and its results (see Table 1), a better system is needed to compensate for the precision. The industry spends a lot of money on the maintenance and purchase of new robotic systems. Some authors have looked at the measurement of precision degradation. It is important to say that despite the advances in robotic vision, there remains a need for more accurate and industry-applicable methods of measuring angular positions in robotic arms. The gap between the existing need and the specific methods that have not yet been implemented in depth are being adsressed.

The methods mentioned in Table 1 are not small enough to meet industry requirements (less than 0.1 millimiters (mm)), or they are so complex that they cannot be applied to other robot morphologies. In this paper, we present a robust angle measurement method that uses one stereo camera, Intel RealSense model D435, and a CNN algorithm to capture the color and depth information of the robot. The red, green, and blue (RGB), i.e., a color space commonly used to color a pixel in images and videos, is useful for segmentation and is mapped to the depth channel with an accuracy of 0.001 mm. The algorithm splits the 3D robot morphology into separate 2D spaces, with one 2D space representing each DoF. The combination of these methods improves the position of the robot with considerable precision. Section 2 describes the materials and methods used in this work. Section 3 shows all the results of this work, Section 4 is the discussion of the findings. Finally, Section 5 refers to the conclusions of this work.

## 2. Materials and Methods

These are the materials and methods used in this work, including the camera and the collaborative robot, also known as a cobot, is an industrial robot that can work safely alongside humans in a shared workspace. This is in addition to the necessary theoretical concepts such as convolutional neural networks, stereoscopic configuration, and others.

### 2.1. Stereo Vision

Stereo vision is a technique for estimating three-dimensional structures of the physical world using two images taken from different angles [6]. Figure 2 shows how the stereo view is composed, which consists of two cameras with the same parameters. At a reference point $P(X_{p},Y_{p},Z_{p})$, the projection onto the right image plane is $p_{1}\left(x_{1},y_{1}\right)$ and onto the left image plane is $p_{2}\left(x_{2},y_{2}\right)$, where *f* is the focal length and *b* is the distance between the optical centers of the two cameras. Referring to the Figure 3 and the principle of similar triangles, we define the disparity *d* in the Equation (Equation 1), then we can obtain the depth *Z* of the point *P* from the Equations (Equation 1) and (Equation 2) [16]:(1)$$d=x_{1}-x_{2},$$
(2)Z=b×f/d,

To estimate the disparity in the digital image pair, the neighborhood of a pixel *p* in the left image is compared to the neighborhood of the pixel image *q* in the right image, where *q* is shifted by a possible disparity $\delta_{p}$ compared to *p*. In Figure 4, for each pixel *p*, *N* possible disparities $\left(\delta_{p}^{1},\delta_{p}^{2},\ldots ,\delta_{p}^{N}\right)$ are tested (N=256 and 65,536 for 8 and 16 bits of depth mapping) and the resulting possible disparity with the lowest fitting cost is assigned to the pixel *p* [33].

#### Accuracy of the D435 Stereo Camera

Using [16,33], the accuracy of the stereo camera was calculated, some images were taken with a ruler on the background to measure 1 centimeter (cm), the distance between two pixels was drawn manually, it was not necessary to obtain 1 cm accuracy. Then, color was added to the two pixels with the information of coordinates (x,y,z). The distance between the pixels was calculated with eq. Then, the same task was performed with the next pixels and the distance between two consecutive pixels was determined, which represents the accuracy of the stereo camera.
(3)$$d=\sqrt{{\left(x_{2}-x_{1}\right)}^{2}+{\left(y_{2}-y_{1}\right)}^{2}+{\left(z_{2}-z_{1}\right)}^{2}},$$

Figure 5 shows the distance between 2 pixels. First, pixels 1 and 2 are measured, then pixels 1 and 3, and so on up to pixel 5. Then, the smallest distance, the distance from pixel 3 minus 2, is calculated, and so on up to pixel 5. The results are shown in Table 2 and Table 3.

As it can be observed, the robot has an accuracy of about 1 mm, which varies a little depending on the distance and the angle of measurement.

### 2.2. Materials

The current work implemented the ISO-9283 [34] for prototype testing. The list of materials and its characteristics used for this work are listed below.

1 MyCobot 280 nano, its main features are:–Workspace 280 mm.–Joint range $\pm 170^{\circ }$.–Repetitively ±5 mm.–A micro-computer Jetson nano 280, with 2 GB of RAM and NVIDIA processor–Servomotors of high precision.–6 DoF.1 depth camera model Intel RealSense D435. Its main features are:–Focal length of 1.88 mm.–Depth resolution 1280×720 pixels, ±0.006 mm–Workspace 0.2–10 m.–6 DoF para calibración.A generic monitor.Mouse and keyboard.

The system is set up on a 150 × 56 cm table, as shown in Figure 6. The Cobot is placed at a distance of 63.5 cm from the top and 32 cm from the left side of the table. The dimensions of the robot are 15 × 11 cm and the camera is 2.5 × 9 cm. The colored circle represents the robot’s working area, while the triangular is the camera’s viewing area. One camera is used at the lower end of the table length and another at the upper end.

As shown in Figure 7, each DoF is measured sequentially from $q_{1}$ to $q_{6}$. Each DoF is enumerated with colored labels to make it easier for the camera to segment them. Also, the dimensions of every link of the Cobot are shown.

### 2.3. Graphical User Interface (GUI)

To enhance the precision when managing cobot manipulation, a Graphical User Interface (GUI), used to create an interaction between the user and the computer, is developed to exhibit the fundamental motions associated with each DoF $\left(\theta_{1},\;\theta_{2},\;\theta_{3},\;\theta_{4},\;\theta_{5},\;\theta_{6}\right)$ and the position values of the end effector $\left(x,\;y,\;z,\;r_{x},\;r_{y},\;r_{z}\right)$. The interface also checks whether the robot is connected and has a button to bring the robot to the home position (all angles to zero) and a button to switch off the servomotors as a precaution. This interface is constantly being improved, so more test modes for positioning will be implemented in the future. Figure 8 shows the GUI.

### 2.4. Measurement of Position Accuracy

Position accuracy is the deviation between the commanded position of the TCP point (N) and the mean (barycenter, G) calculated from the cluster of repeatedly reached TCP positions. Figure 9 shows the position accuracy of 1 DoF, where G is the barycentre $\left[\bar{x},\bar{y},\bar{z}\right]$, and N is the commended position $\left[x,y,z\right]$. The position accuracy of 1 DoF $a_{p}$ is given by Equation (Equation 4).
(4)$$a_{p}=\sqrt{a_{px}^{2}+a_{py}^{2}+a_{pz}^{2}},$$
where $a_{px}=\bar{x}-x_{cmd}$, $a_{py}=\bar{y}-y_{cmd}$, and $a_{pz}=\bar{z}-z_{cmd}$. The position accuracy $a_{p}$ of a robot manipulator is defined as the distance between the commanded pose $x_{cmd}$ and the mean $\bar{x}$ of *N* reached positions, as described in Equation (Equation 5)
(5)$$x_{cmd}=\left[\begin{array}{c} x_{cmd}\\ y_{cmd}\\ z_{cmd} \end{array}\right],\bar{x}=\left[\begin{array}{c} \bar{x}\\ \bar{y}\\ \bar{z} \end{array}\right],$$

The mean coordinates are given in Equation (Equation 6).
(6)$$\bar{x}=\frac{1}{N}\sum_{i=1}^{N}x_{i},\bar{y}=\frac{1}{N}\sum_{i=1}^{N}y_{i},\bar{z}=\frac{1}{N}\sum_{i=1}^{N}z_{i},$$

$x_{i}$, $y_{i}$, and $z_{i}$ are the position coordinates of the *i*-th measured pose. The positioning accuracy $a_{p}$ is obtained using the Euclidean norm (Equation 7).
(7)$$a_{p}=∥\left(\bar{x}-{\bar{x}}_{cmd}\right)∥=\sqrt{{\left(\bar{x}-x_{cmd}\right)}^{2}+{\left(\bar{y}-y_{cmd}\right)}^{2}+{\left(\bar{z}-z_{cmd}\right)}^{2}},$$

A measurement plan is required to control the robot movements. The controlled robot movements for the evaluation of accuracy degradation cannot be arbitrary. The measurement plan requires that the robot TCP moves over the entire working area and is evenly distributed in both articulated and Cartesian space. The even distribution of the samples prevents the analysis algorithm from overlooking or overweighting errors, which would falsify the results. Covering the entire joint space and Cartesian space means that the measurement plan trains the robot over a subset of joints or work zones. This allows the performance of the joints to be recorded over the entire motor and encoder range. Covering the entire workspace helps to evaluate different stiffness conditions. When creating the measurement plan, a collision check is performed to minimize possible interruptions during movement and measurement. A visibility check is also performed to ensure that the planned positions are not obscured by the measuring instrument [2].

A Cobot 280 Nano from Elephant Robotics was used for this work. The fixed-loop motion generated for the cobot is shown in Figure 10. As the cobot was mounted on a table, only poses above the table are valid for this use case. The robot movement assumes an unobstructed working volume above the table.

Based on [2], a grid moment of the Cobot 280 Nano is planned. The tool center positions (TCP) (*x*, *y*, *z*) of the robot are measured. The differences between the nominal positions and the measured positions are compared. Control-level measurement data are also collected from each joint to understand the influences of temperature, payload, and velocity on position changes. Control-level information provides clues about the causes of robot performance degradation by providing information on the actual and joint positions, velocities, currents, accelerations, torques, and temperatures.

### 2.5. Convolutional Neural Networks

Also known as CNNs, these are the standard neural network architecture used for predictions when the input are images, which is the case in a variety of neural network applications, such as this work.

The implemented architectures for this projects are the followed:

**LeNet-5** is one of the earliest pre-trained models proposed by Yann LeCun and others in 1998 in the research paper Gradient-Based Learning Applied to Document Recognition. They used this architecture to recognize handwritten and machine-printed characters. The main reason for the popularity of this model was its simple architecture. It is a multilayer convolutional neural network for image classification, Figure 11 shows the description of its architecture [35].

**AlexNet** is a convolutional neural network model proposed by K. Simonyan and A. Zisserman. The model achieves a top-5 test accuracy of 92.7% on ImageNet, a dataset with over 14 million images belonging to 1000 classes, Figure 12 displays the architecture of this network [36].

**YOLO−v1** “You Only Look Once” is a system for recognizing objects on the Pascal VOC 2012 dataset. It can recognize the 20 Pascal object classes (most of them are animals). All previous recognition systems use classifiers or localizers to perform recognition. They apply the model to an image at multiple locations and scales. Regions with high scores are considered recognitions. The YOLO designers use a completely different approach. We apply a single neural network to the entire image. This network divides the image into regions and predicts bounding boxes and probabilities for each region. These bounding boxes are weighted according to the predicted probabilities [37].

Using all the methods and materials presented in this section, the results achieved are presented below.

## 3. Results

First, inverse kinematics was obtained using CNN, as a first test of the Deep Learning techniques, the angle of each DoF $\left(\theta_{1},\;\theta_{2},\;\cdots ,\;\theta_{6}\right)$ and the geometric position of the TCP $\left(x,\;y,\;z\right)$ were recorded and stored to run the algorithm. The robot moved into its workspace with 16 different positions or setpoints; finally, a matrix of 3070×9 data was created. Figure 13 shows the position of every axis (*x, y, z*), the setpoint and actual value of every one, as it can be observed precision is observable in the graphs the fourth graph is all the points in a three-dimensional plot.

The algorithm for predicting the inverse kinematic solution is a CNN of 1 dimension that takes the TCP points (*x, y, z*) as the inputs and the angles of the DoF as the outputs $\left(\theta_{1},\;\theta_{2},\;\cdots ,\;\theta_{6}\right)$. The results show that this is a powerful task, traditional methods are more complicated. Figure 14 shows the model of the one-dimensional CNN developed for this task, and Figure 15 shows the training results of the CNN. For this test, 3024 samples were used, with 2456 for training and 614 for validation. The Adam optimizer was used, including the mean squared error loss, and the accuracy metric was plotted.

The next step in testing the DL techniques was to use the depth information of the cobot image to predict the results of the inverse kinematics. The D435 stereo camera provides color and depth information, 680×460 pixels for the RGB image, and the depth information has the same resolution. Thus, each color pixel can be represented in a depth pixel with a 16-bit value.

A sample of RGB images of the robot can be found in Figure 16. Figure 17 shows the same images but with depth information. For training this model, 120 images were used. The information of the angle of each DoF was replaced by a depth image of the robot, which makes the design of the CNN more complicated. LENET-5 and AlexNet designs were implemented.

Figure 18 shows the results of the training of LENET-5 and Figure 19 for AlexNet. For these tests 160 samples were used, 128 for training and 32 for validation. For LENET-5, the Adam optimizer was used, including the mean squared error loss and the accuracy metric. AlexNet was configured with the AdamMax optimizer.

The next steps were to put the colored labels on the first four DoF of the cobot (4 out of 6 DoF were used) and train a network to recognize the labels through learning; this is shown in Figure 20. A blue, green, yellow, and red label for DoF 1 to 4, respectively, was attached to a specific point on the robot. The label was rounded and had a radius of 0.6 cm.

For label annotation, the CVAT platform was used, it allowed for converting raw images into a custom trained computer vision model. It supports object recognition and classification models.

For label recognition, YOLO-V1 was implemented, running in the Roboflow platform. The results are shown in Figure 21. For this task, 215 images of the robot in different posses were implemented. The algorithm divided the images into 151 images for training, 42 for validation and 22 for testing.

Figure 22 shows some images of the Cobot that were classified, none of them were used for training, validation, or testing stage, the results given are satisfactory, the next step in the experimentation is to interpolate labels information with depth information of an image to locate the tridimentional position of every DoF.

Depth camera D435 provided the tridimensional information *(x, y, z)* of every pixel. Figure 23 shows the depth information, a similar set of data with x and y orientation was also obtained.

Yolo-V1 provided information for every label in the Cobot, information can be interpolated to the depth image so we can obtain the tridimensional information of every DoF. This can be observed in Figure 24.

This methodology outperforms all work with an average error of 1 mm. Moreover, the set of our implemented methods leads to clearer and reproducible work. It is the first time that stereo cameras and CNN are used to solve this particular problem.

## 4. Discussion

In this work, a measurement position with an accuracy of about 1 mm was achieved (limited by the standard camera resolution of 1 mm). Compared with previous work, conventional techniques are more accurate but more difficult to reproduce as they involve geometric analysis. Other previous works that present an intelligent algorithm and/or robot vision have an error of less than 1 mm and above. Otherwise, other works do not indicate the degree of improvement. Our work presents a methodology to measure the degradation of the positioning of the links of a Cobot, using a combination of artificial intelligence and computer vision. This combination makes it possible to obtain results that are reproducible in other architectures of a manipulator robot. It is important to mention that the reduction of the error in the measurement of the link position can be reduced by using a more precise camera and basically the same methodology presented. This opens up a line of work that can benefit the industry by extending the life of a robot while ensuring the quality of a product [2]. Based on our results, we should consider technologies with higher measurement accuracy in the future, such as a KEA camera with Time-of-Flight (ToF) technology, or replacing the depth technology with a lidar system. As mentioned earlier, there were certain limitations in terms of the resolution of the camera, which is the main drawback of this work. This system is suitable for applications where an accuracy of 1 mm is required and where the integration of non-invasive sensors such as cameras is necessary. However, it is necessary to work on the use of two chambers to avoid occlusions.

## 5. Conclusions

With the implemented methods, convolutional neural networks and stereo depth cameras, there is an intuitive adaptation that makes experimentation relatively comfortable. Preliminary results show that the camera tends to have a precision of 1 mm resolution, which is not sufficient for our application. However, in the future, we will work with newer stereo cameras or other depth technology. Cobot is a good tool to work with as its accuracy is in the millimeter range, which is an important requirement for our work. LENET-5 and AlexNet as well as a proprietary CNN are used to first determine the inverse kinematics of the robot, initially with pure data from the encoders, then depth images are included with good learning results, for labeling with YOLO V1 the accuracy was quite acceptable with 91%. Finally, the methods used to analyze our positioning improvement are the most appropriate and advanced. Compared with previous methods, our results are more reproducible, with a maximum resolution of 1 mm. Depth cameras bring us closer to better results, although not the best, which is why we intend to use higher resolution stereo cameras. These deep vision methods could be the standard of the future.

## Figures and Tables

**Figure 1 biomimetics-09-00610-f001:**
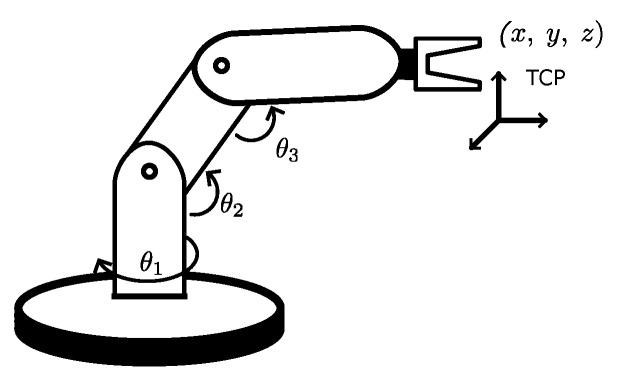
A 3-DoF robot architecture and its principal components, this model is simplified to understand the main concepts [1].

**Figure 2 biomimetics-09-00610-f002:**
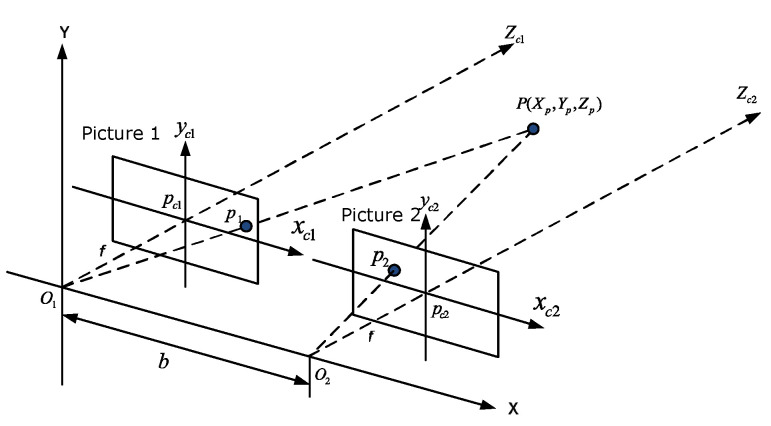
Schematic of stereo vision configuration in three dimensions [16,32].

**Figure 3 biomimetics-09-00610-f003:**
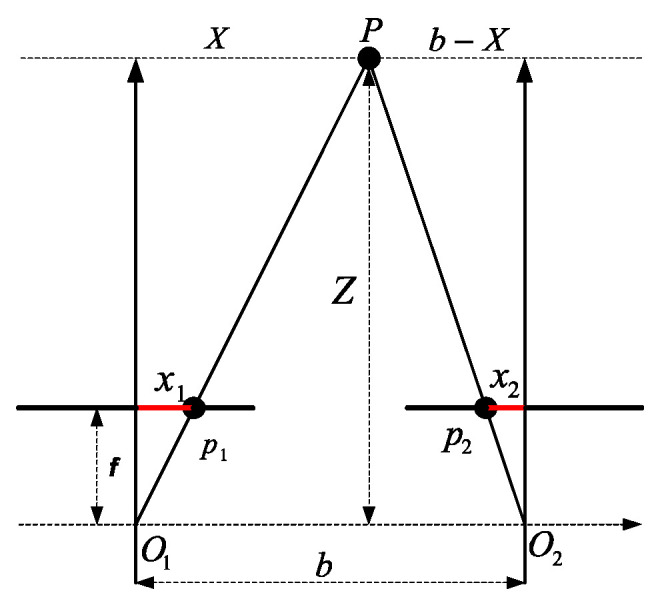
Triangulation scheme analysis for stereo vision a two dimensional perspective [16].

**Figure 4 biomimetics-09-00610-f004:**
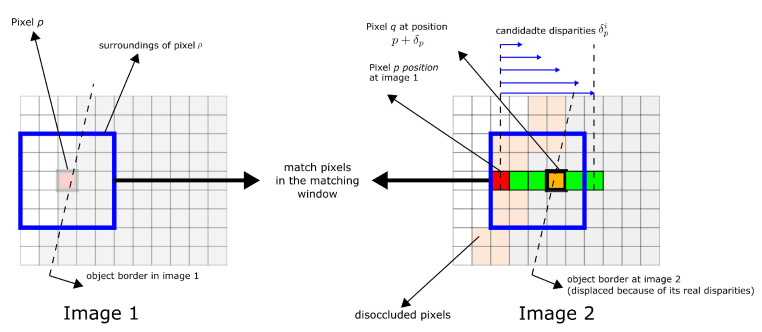
Matching windows in stereo vision. This represents the digital functionality of a commercial stereo camera [33].

**Figure 5 biomimetics-09-00610-f005:**
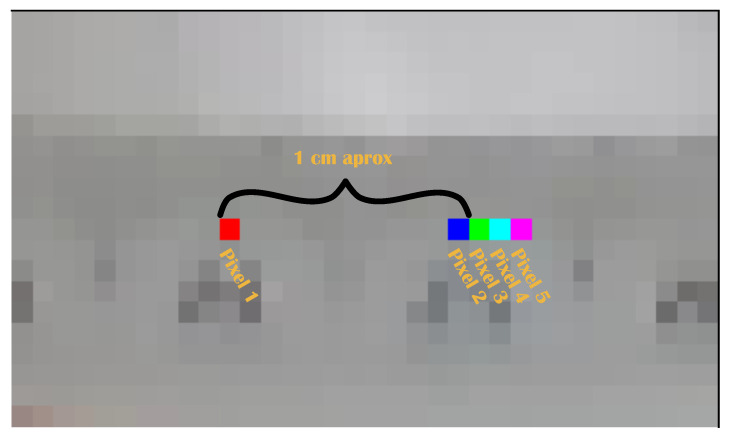
Distance of 1 cm in pixels, and the subsequent pixel to measure the D435 camera.

**Figure 6 biomimetics-09-00610-f006:**
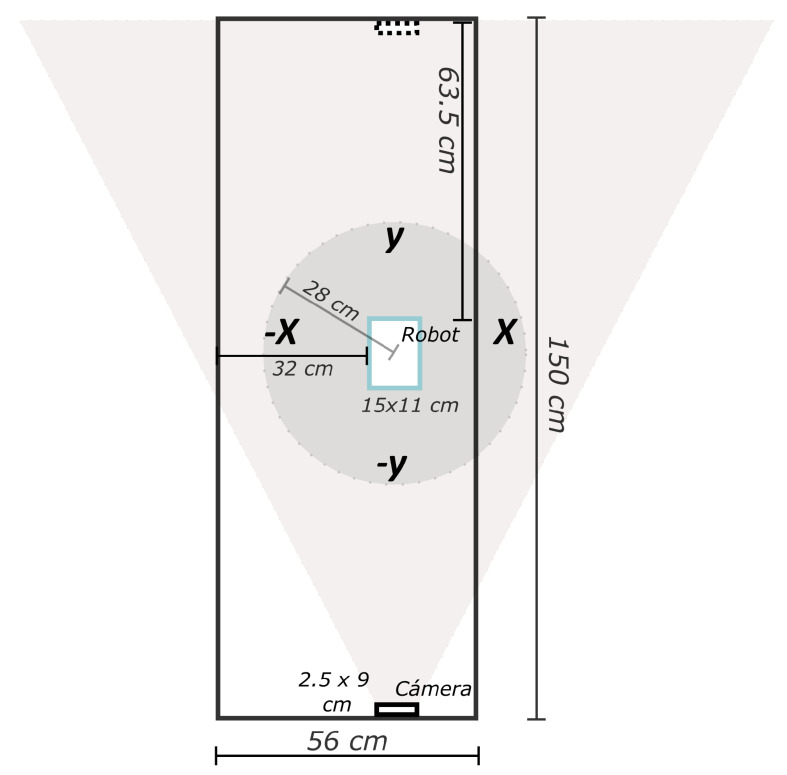
Configuration of the materials, the rectangle represents a table of 56 × 150 cm, the triangle is the view of the stereo camera, the circle is the workspace of the robot.

**Figure 7 biomimetics-09-00610-f007:**
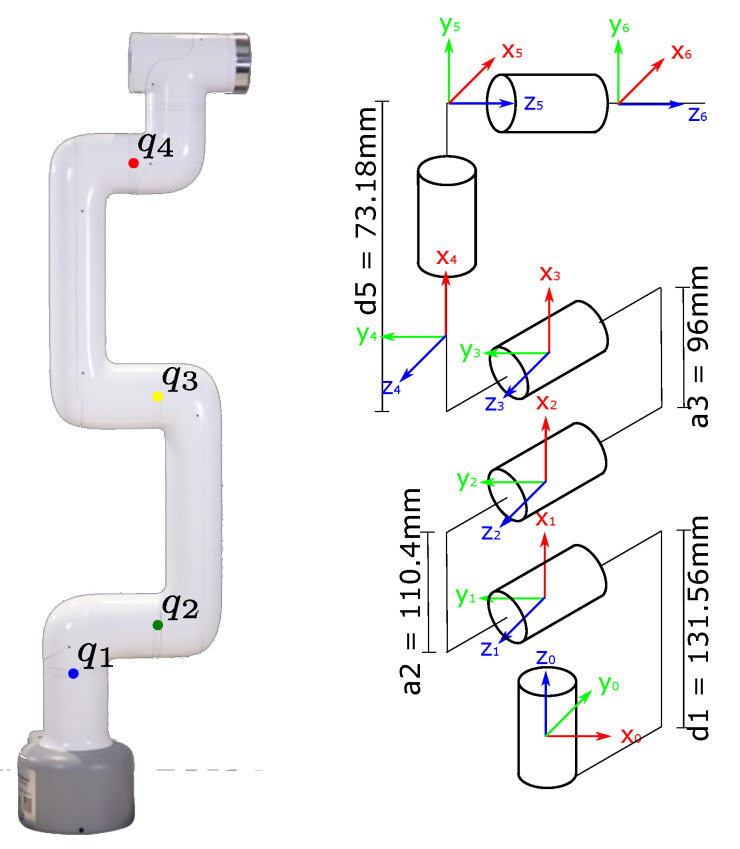
6 DoF Cobot morphology. Only 4 DoF are labeled with a different color.

**Figure 8 biomimetics-09-00610-f008:**
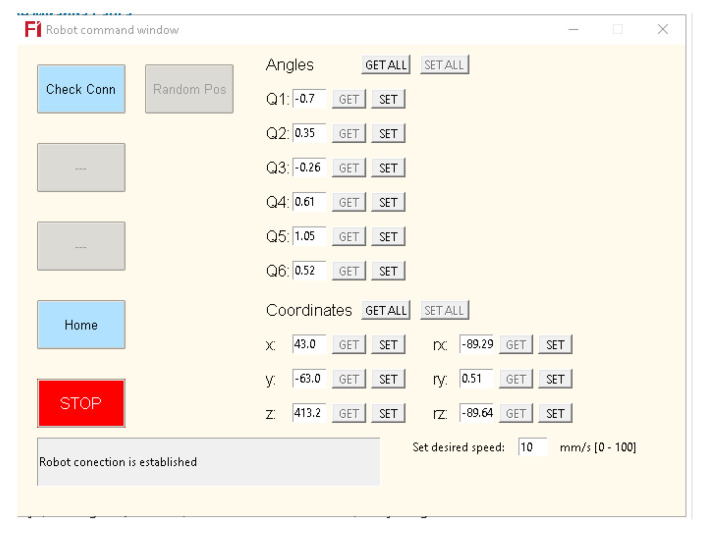
GUI used for better manipulation of the robot arm, for communication; RPC (Remote Procedure Call) protocol is implemented, RPC communication is based on TCP (Transfer Control Protocol) architecture.

**Figure 9 biomimetics-09-00610-f009:**
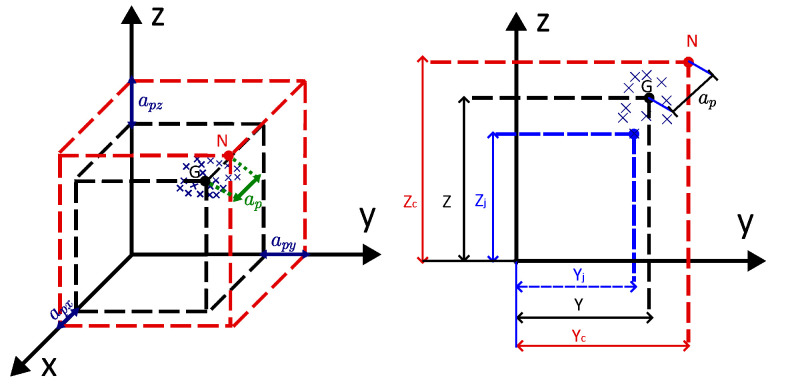
Graphical displaying of 1 DoF position accuracy according to ISO 9283-1998 [34].

**Figure 10 biomimetics-09-00610-f010:**
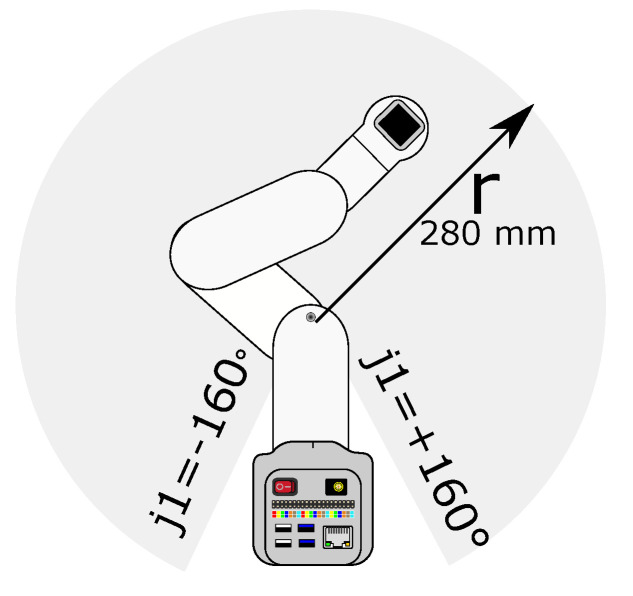
Workspace of the Cobot 280 Nano.

**Figure 11 biomimetics-09-00610-f011:**
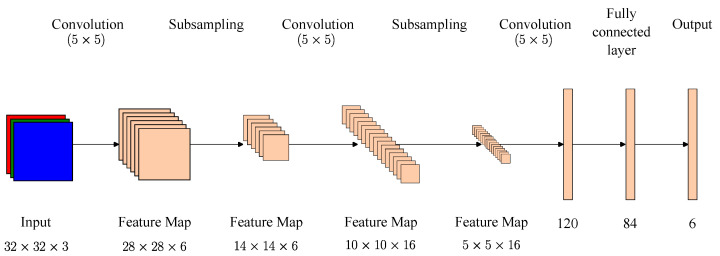
LENET-5 architecture [35].

**Figure 12 biomimetics-09-00610-f012:**
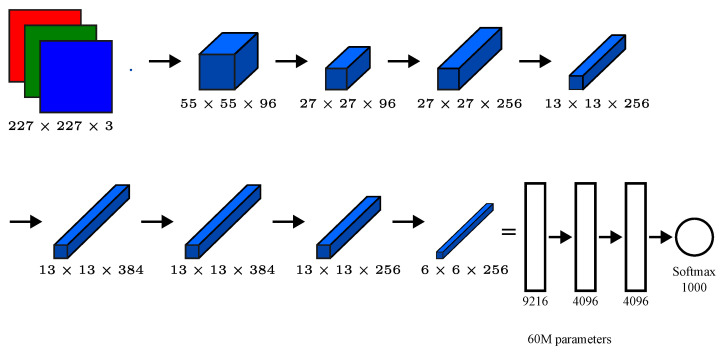
AlexNet architecture [36].

**Figure 13 biomimetics-09-00610-f013:**
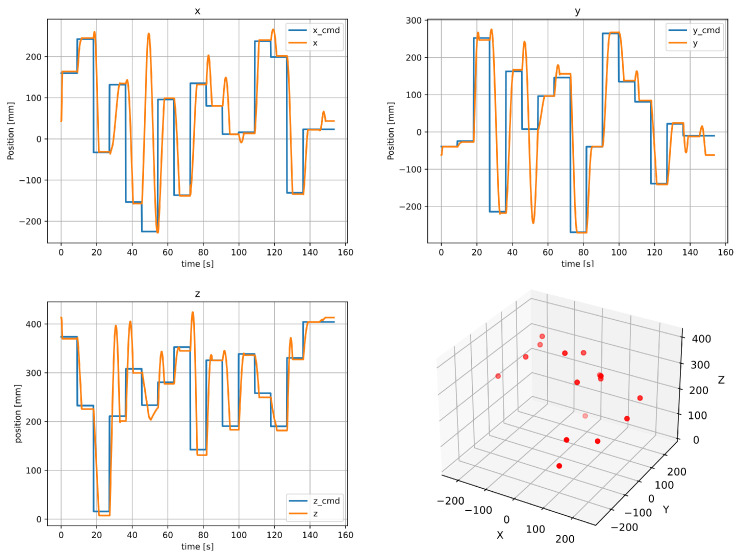
Every three-dimensional position of TCP position (*x, y, z*), the fourth graph is the points that the Cobot follows.

**Figure 14 biomimetics-09-00610-f014:**
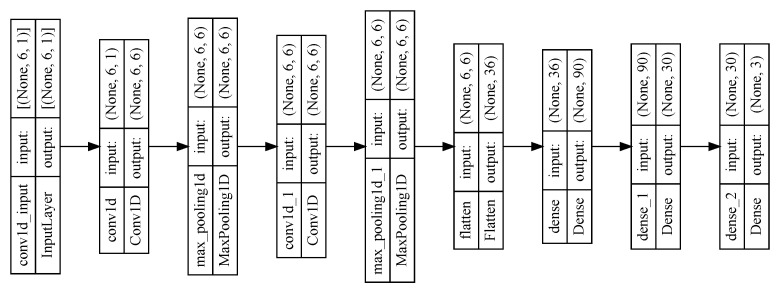
Design of the 1-dimensional CNN used for predicting inverse kinematics, highlighting the input layer, convolutional layers, and output.

**Figure 15 biomimetics-09-00610-f015:**
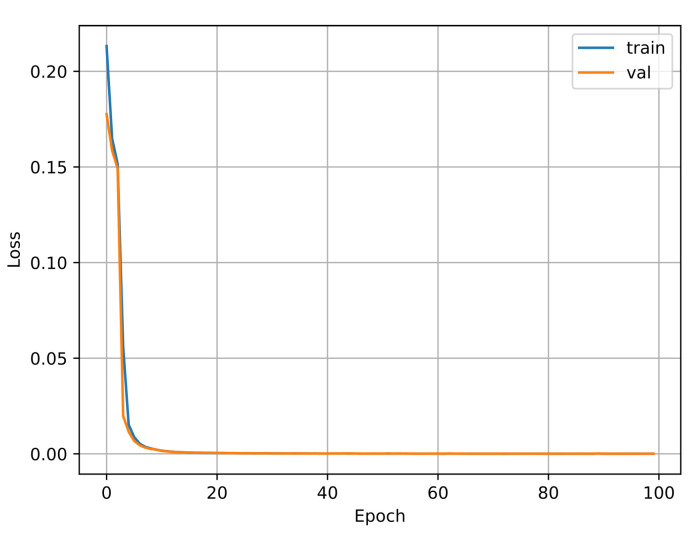
Training vs. validation loss function of the 1 dimensional CNN.

**Figure 16 biomimetics-09-00610-f016:**
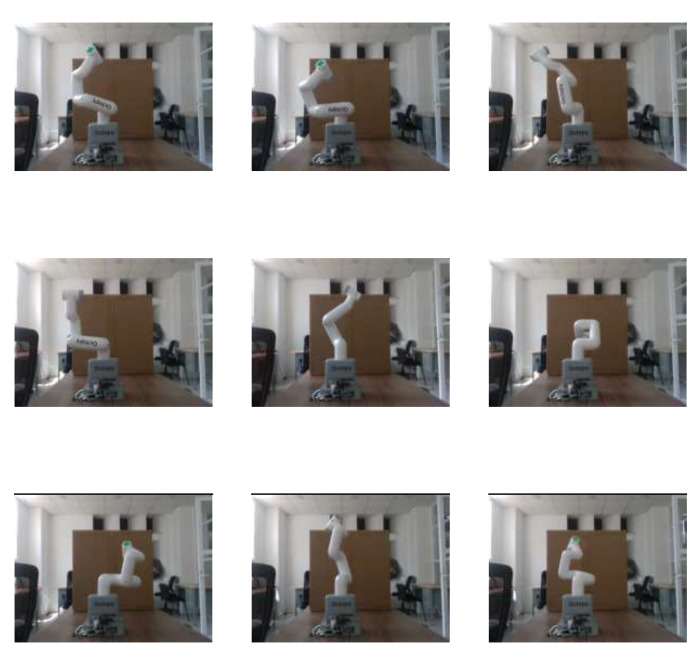
Nine samples of RGB pictures of the Cobot, the pixels are 480 × 640 pixels.

**Figure 17 biomimetics-09-00610-f017:**
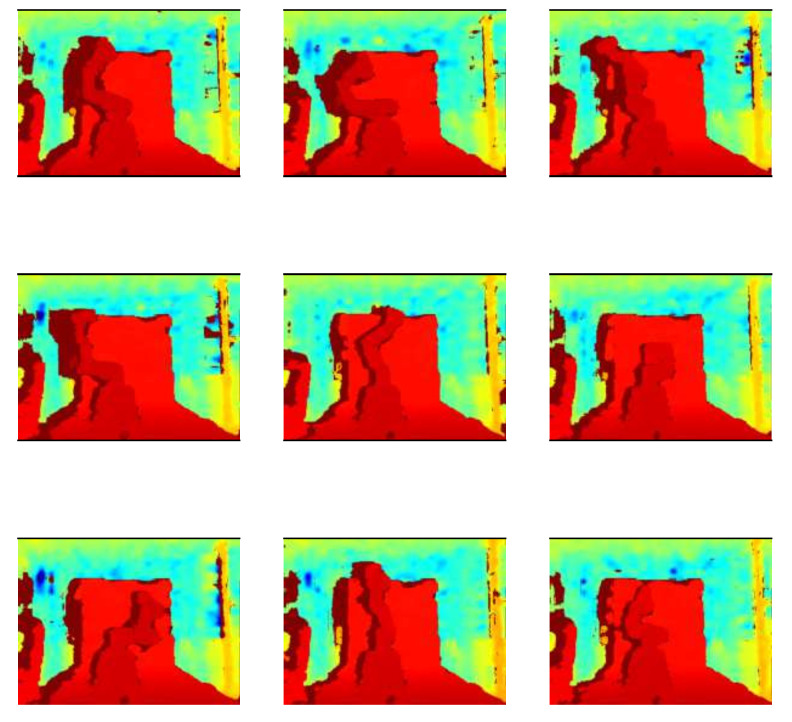
Nine samples of depth information for the Cobot, it has the same size of a color image, every pixel has a resolution of 16 bits.

**Figure 18 biomimetics-09-00610-f018:**
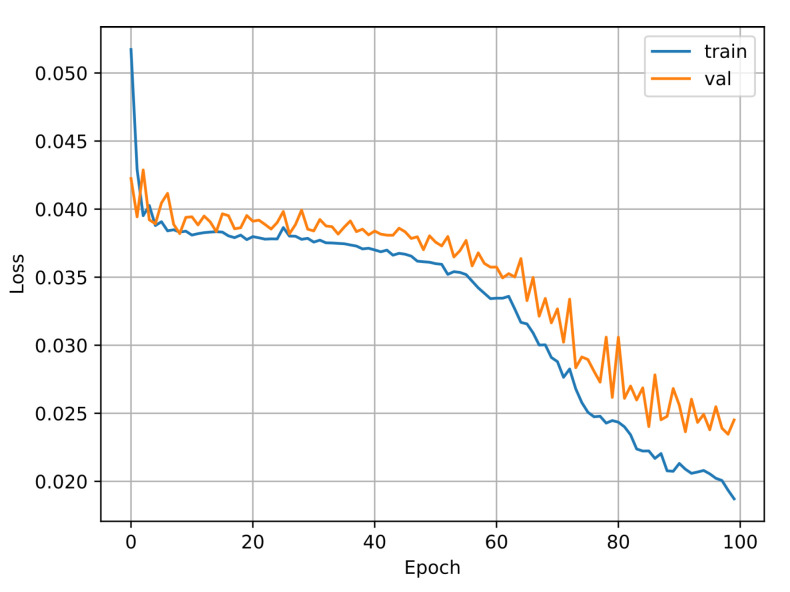
Training vs. validation loss function of LENET-5.

**Figure 19 biomimetics-09-00610-f019:**
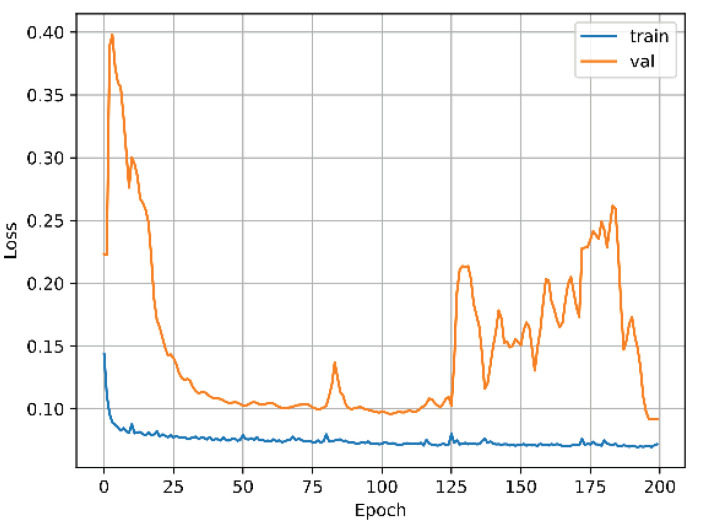
Training vs. validation loss function of AlexNet.

**Figure 20 biomimetics-09-00610-f020:**
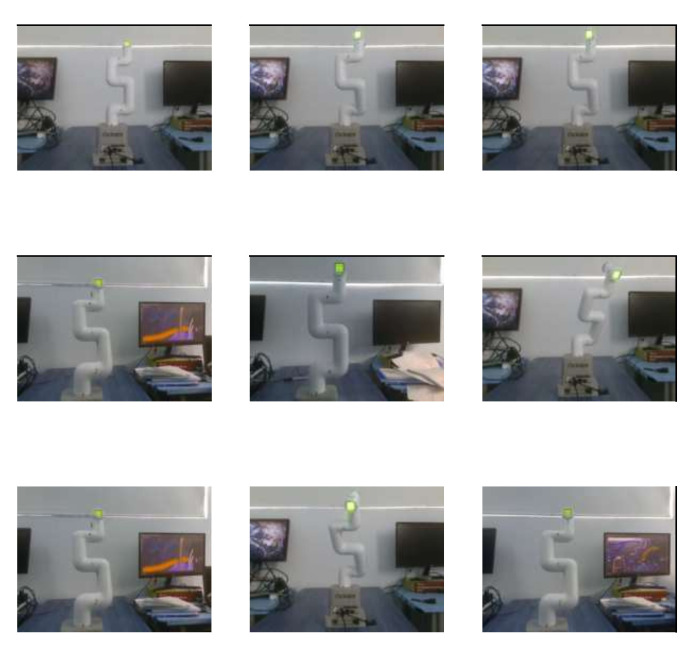
Images of the robot with colored labels, using YOLO-V1.

**Figure 21 biomimetics-09-00610-f021:**
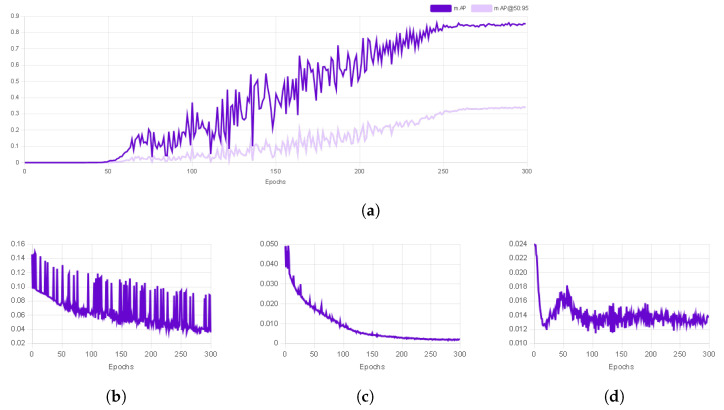
Yolo-V1 training results for label identification. (**a**) **Mean average precision (mAP)** is used to measure the performance of computer vision models. (**b**) **Box Loss** refers to how well the model predicts the positions and sizes of bounding boxes around objects in an image. (**c**) **Class Loss** refers to the measure of how accurately the model is predicting the correct class or label of the objects it detects. (**d**) **Object Loss** measures how well the model recognizes the presence or absence of an object in a particular region of an image.

**Figure 22 biomimetics-09-00610-f022:**
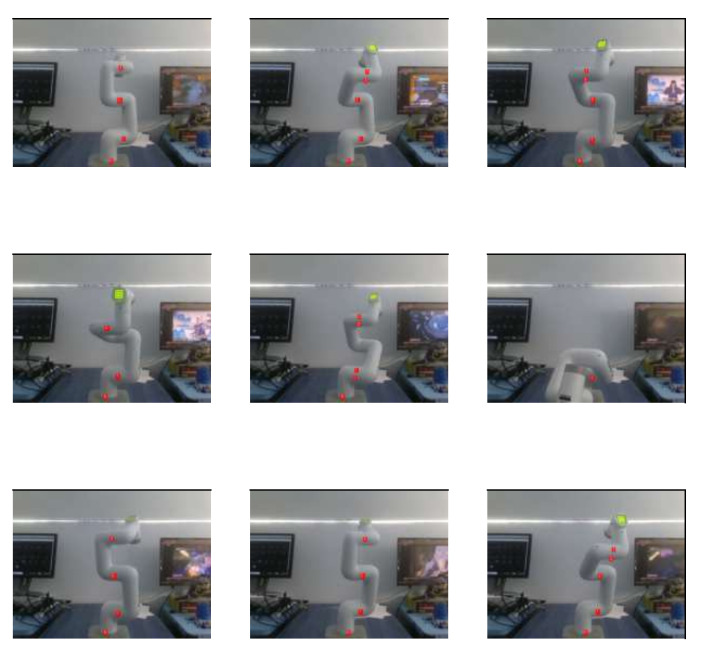
Results with a new set of images to evaluate the Yolo-V1 algorithm. The red dots are the recognized labels.

**Figure 23 biomimetics-09-00610-f023:**
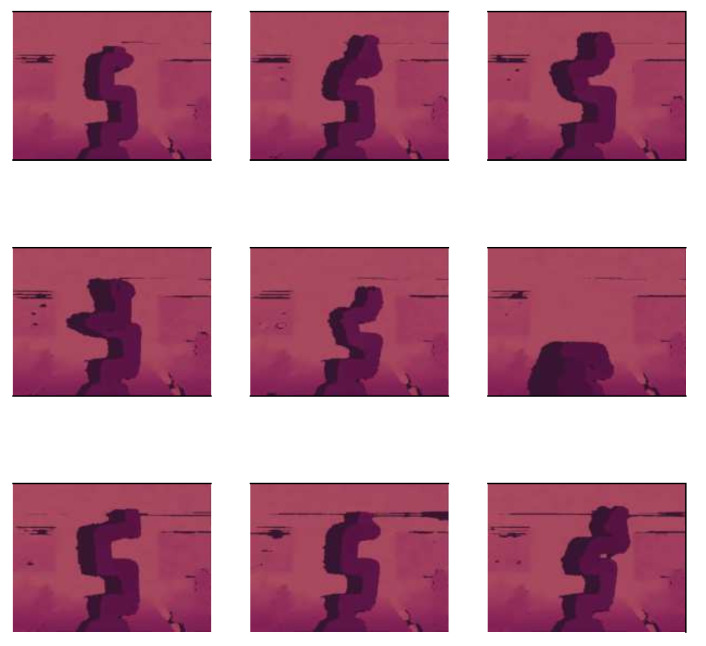
Depth information of the color images.

**Figure 24 biomimetics-09-00610-f024:**
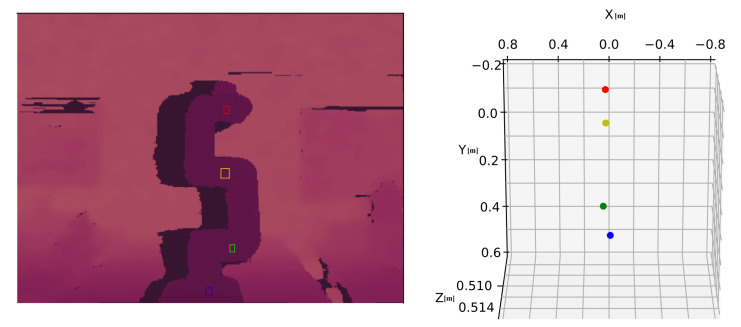
The image on the left is the depth information of the color images with the location of every label. The color box represents the workspace where labels are projected. The dots in the color box are the marked points detected by the YOLO-V1 algorithm that helps measure each DoF angular position, where blue dot is the first DoF label, green dot is the second DoF label, yellow dot is the third DoF label and finally, the red dot is the fourth DoF label.

**Table 1 biomimetics-09-00610-t001:** Summary of state-of-the-art. The methodology implemented in every work was checked with a ✓.

Reference	Conventional Techniques	AI Techniques	Robotic Vision	ME before [mm]	ME after [mm]
[23]	✓				
[24]	✓	✓		2.613	0.31
[25]	✓	✓		0.7411	0.1007
[26]		✓	✓		27.4
[27]		✓	✓		
[28]	✓			1.2248	0.2678
[29]	✓	✓			0.1
[30]	✓	✓			
[31]	✓	✓			

**Table 2 biomimetics-09-00610-t002:** Distance between pixels for measuring the accuracy of the camera according to Figure 5. The distance between pixel 1 and the following consecutive pixels.

Pixels	Distance [mm]
1 to 2	13.9
1 to 3	15.0
1 to 4	16.3
1 to 5	17.7

**Table 3 biomimetics-09-00610-t003:** The distance between two consecutives pixels is the resolution for distance measurement.

Pixels	Distance between Two Consecutive Pixels [mm]
3–2	1.1
4–3	1.3
5–4	1.4

## Data Availability

The original contributions presented in the study are available on request for the corresponding author.

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
