# Peer review of "Pose Estimation of a Cobot Implemented on a Small AI-Powered Computing System and a Stereo Camera for Precision Evaluation"

_biomimetics, 2024, doi:10.3390/biomimetics9100610_

Round 1

Reviewer 1 Report

Comments and Suggestions for Authors

1. Introduction

  • Comment: The introduction provides a solid overview of the importance of precision in robotic manipulators, but it could benefit from a more explicit statement of the research gap.
  • Suggestion: Consider adding a sentence that clearly identifies the specific gap in the literature that this research addresses. For example: "Despite advancements in robotic vision techniques, there remains a need for more accurate and industry-applicable methods for measuring angular positions in robotic arms."

2. Clarity and Conciseness

  • Comment: Some sentences are overly complex, making the content harder to digest.
  • Suggestion: Simplify sentences where possible. For example, "This method does not pretend to replace encoders, but to compensate the degradation of accuracy through an intelligent and visual measurement system" could be rephrased as "This method is not intended to replace encoders but to enhance accuracy by compensating for degradation through an intelligent visual measurement system."

3. Technical Terminology

  • Comment: The use of technical terms is appropriate, but some terms may benefit from brief explanations for readers who may not be experts in all aspects of the field.
  • Suggestion: When first introducing terms like "CNN" or "stereo vision," consider briefly explaining them, even if they are commonly understood in your field. For instance: "Convolutional Neural Networks (CNNs), a type of deep learning model particularly effective for image analysis, were used..."

4. Consistency

  • Comment: There are slight inconsistencies in the use of abbreviations and units.
  • Suggestion: Ensure that terms like "DoF" (Degrees of Freedom) and units like millimeters (mm) are used consistently throughout the document. Once an abbreviation is introduced, use it consistently thereafter.

5. Figures and Tables

  • Comment: The figures and tables are helpful but could benefit from more descriptive captions.
  • Suggestion: Enhance the captions to be more descriptive, explaining what the reader should observe in the figure or what key information the table provides. For example, instead of just "Figure 14. Design of 1 dimensional CNN," consider "Figure 14. Design of the 1-dimensional CNN used for predicting inverse kinematics, highlighting the input layer, convolutional layers, and output."

6. Punctuation and Grammar

  • Comment: The grammar is generally good, but there are minor punctuation errors that could be corrected for better readability.
  • Suggestion: Review the document for comma usage, particularly in complex sentences. For example, "For this test, 3024 samples were implemented, 2456 for training and 614 for validation." could be restructured for clarity: "For this test, 3024 samples were used, with 2456 for training and 614 for validation."

7. Discussion and Conclusion

  • Comment: The discussion and conclusion sections are somewhat brief and could provide a deeper analysis of the results in relation to previous studies.
  • Suggestion: Expand on how your results compare with previous research mentioned earlier in the document. Discuss the implications of your findings in greater detail, particularly how they advance the field or open up new avenues for future research.

8. Future Work

  • Comment: The mention of future work is positive, but it could be more specific.
  • Suggestion: Provide more concrete examples of how you plan to address the limitations of your current study in future research. For example, you might specify what newer stereo cameras you intend to use and what specific improvements you expect them to bring.

9. References

  • Comment: The references are comprehensive but should be consistently formatted.
  • Suggestion: Double-check the reference formatting to ensure consistency with the journal's requirements. Make sure all references are cited in the correct style and that URLs and DOIs are properly formatted.

10. Overall Structure

  • Comment: The structure is logical, but transitions between sections can be improved to enhance the flow of the document.
  • Suggestion: Use transitional sentences at the beginning and end of each section to guide the reader through your argument. For example, after presenting results, you might lead into the discussion by stating, "These results suggest a significant improvement in robotic arm accuracy, which we will now discuss in the context of existing methods."
Comments on the Quality of English Language

The English is of good quality and appropriate for an academic or technical audience. Still, there are opportunities to refine the language for better clarity, consistency, and readability.

Author Response

REVIEWER 1

Thank you very much for taking the time to review this manuscript. Please find the detailed responses below and the corresponding revisions/corrections highlighted/in track changes in the re-submitted files. The changes are highlighted in red.

Comment 1: The introduction provides a solid overview of the importance of precision in robotic manipulators, but it could benefit from a more explicit statement of the research gap.

Suggestion: Consider adding a sentence that clearly identifies the specific gap in the literature that this research addresses. For example: "Despite advancements in robotic vision techniques, there remains a need for more accurate and industry-applicable methods for measuring angular positions in robotic arms."

Response 1: Thank you for your suggestion. The comment has been addressed and the suggested changes are included in the new version of the manuscript.

Comment 2: Some sentences are overly complex, making the content harder to digest.

Suggestion: Simplify sentences where possible. For example, "This method does not pretend to replace encoders, but to compensate the degradation of accuracy through an intelligent and visual measurement system" could be rephrased as "This method is not intended to replace encoders but to enhance accuracy by compensating for degradation through an intelligent visual measurement system."

Response 2: We appreciate the comment, as well as the suggestion to reduce the complexity of some sentences. A revision has been made throughout the article to simplify some sentences.

Comment 3: The use of technical terms is appropriate, but some terms may benefit from brief explanations for readers who may not be experts in all aspects of the field.

Suggestion: When first introducing terms like "CNN" or "stereo vision," consider briefly explaining them, even if they are commonly understood in your field. For instance: "Convolutional Neural Networks (CNNs), a type of deep learning model particularly effective for image analysis, were used..."

Response 3: We appreciate your comments and suggestions. We have made the suggested adjustments to the text of the article, adding a brief explanation of the technical terms used.

Comment 4: There are slight inconsistencies in the use of abbreviations and units.

Suggestion: Ensure that terms like "DoF" (Degrees of Freedom) and units like millimeters (mm) are used consistently throughout the document. Once an abbreviation is introduced, use it consistently thereafter.

Response 4: Thank you for your time and comments. A search for inconsistencies in abbreviations and units has been carried out and corrected according to the suggestions made by the reviewer.

Comment 5: The figures and tables are helpful but could benefit from more descriptive captions.

Suggestion: Enhance the captions to be more descriptive, explaining what the reader should observe in the figure or what key information the table provides. For example, instead of just "Figure 14. Design of 1 dimensional CNN," consider "Figure 14. Design of the 1-dimensional CNN used for predicting inverse kinematics, highlighting the input layer, convolutional layers, and output."

Response 5: Thank you for your comments and suggestions. We have revised the figures caption, and the explanation has been extended, especially in those where there were few words.

Comment 6: The grammar is generally good, but there are minor punctuation errors that could be corrected for better readability.

Suggestion: Review the document for comma usage, particularly in complex sentences. For example, "For this test, 3024 samples were implemented, 2456 for training and 614 for validation." could be restructured for clarity: "For this test, 3024 samples were used, with 2456 for training and 614 for validation."

Response 6: Thank you for your time, comments and suggestions. A revision has been made to reduce the complexity of some sentences, adding or mitigating a little bit the sentence, throughout the article.

Comment 7: The discussion and conclusion sections are somewhat brief and could provide a deeper analysis of the results in relation to previous studies.

Suggestion: Expand on how your results compare with previous research mentioned earlier in the document. Discuss the implications of your findings in greater detail, particularly how they advance the field or open up new avenues for future research.

Response 7: Thank you for your comments and suggestions. The discussion and conclusions sections have been extended, and mention is made of the implications of what has been presented, as well as the impact it may have on future research.

Comment 8: The mention of future work is positive, but it could be more specific.

Suggestion: Provide more concrete examples of how you plan to address the limitations of your current study in future research. For example, you might specify what newer stereo cameras you intend to use and what specific improvements you expect them to bring.

Response 8: Thank you for your comments and suggestions. In future work, the information has been extended, considering the current limitations of the work, and what remains to be done in the future.

Comment 9: The references are comprehensive but should be consistently formatted.

Suggestion: Double-check the reference formatting to ensure consistency with the journal's requirements. Make sure all references are cited in the correct style and that URLs and DOIs are properly formatted.

Response 9: Thankyou for your comment and suggestion. The references are made automatically with the latex editor, these references are downloaded from the official pages of the journals in bibtex format and are converted to bibtem format.

Comment 10: The structure is logical, but transitions between sections can be improved to enhance the flow of the document.

Suggestion: Use transitional sentences at the beginning and end of each section to guide the reader through your argument. For example, after presenting results, you might lead into the discussion by stating, "These results suggest a significant improvement in robotic arm accuracy, which we will now discuss in the context of existing methods."

Response 10: We appreciate the time spent on the review, as well as the comments and suggestions made. A brief transition comment was added at the end of each section to improve the reading flow of the document.

Reviewer 2 Report

Comments and Suggestions for Authors

Please see the attached reviewer report.

Comments on the Quality of English Language

Extensive editing of English language required.

Author Response

RESPONSES FOR THE REVIEWER 2

Thank you very much for taking the time to review this manuscript. Please find the detailed responses below and the corresponding revisions/corrections highlighted/in track changes in the re-submitted files. The changes are highlighted in red.

Comment 1: Ensure the end of each line of the equations has a punctuation, either a comma or a full stop if it is the end of the equation - these are missing in several of the equations.

Response 1: Thank you for your comments. The equations were corrected by adding the comma at the end of each one, and this can be seen in the new version of the manuscript.

Comment 2: In the abstract present tense is more suitable instead of past tense.

Response 2: Thank you for your suggestion. The comment has been addressed and the abstract has been changed to present tense.

Comment 3: The Introduction part has been stated properly but, it should be improved by considering the related literature and should point out the motivation to the paper. It should make a compelling case for why the study is useful along with a clear statement of its novelty or originality by providing relevant information and providing answers to basic questions.

Response 3: We appreciate your valuable suggestion. The comment was addressed by adding literature related to the topic of this work. In addition, the motivation for the development of this work was included.  The novelty of the work is also highlighted.

Comment 4: The paper has several typos and grammatical errors. It must be reconsidered the whole paper in terms of the language and typo.

Response 4: We apologize for the grammatical errors and typos. We would like to mention that the manuscript has been completely reviewed and the errors have been corrected.

Comment 5: The authors should revise the Discussion part which is not related to the paper!!!

Response 5: We apologies for the inconvenience. By mistake we load a previous version of the manuscript. The correct version of the manuscript includes the correct discussion section.

Round 2

Reviewer 2 Report

Comments and Suggestions for Authors

After reading the revised form of the manuscript, I am convinced that the authors have improved their previous version making it now suitable for acceptance by the Biomimetics (ISSN 2313-7673) journal.

Comments on the Quality of English Language

Minor errors that should be corrected.